# Enhancement of Complex Permittivity and Attenuation Properties of Recycled Hematite (α-Fe_2_O_3_) Using Nanoparticles Prepared via Ball Milling Technique

**DOI:** 10.3390/ma12101696

**Published:** 2019-05-24

**Authors:** Ebenezer Ekow Mensah, Zulkifly Abbas, Raba’ah Syahidah Azis, Ahmad Mamoun Khamis

**Affiliations:** 1Department of Physics, Faculty of Science, Universiti Putra Malaysia, Serdang 43400, Malaysia; ebemensek@yahoo.co.uk (E.E.M.); rabaah@upm.edu.my (R.S.A.); akhameis@yahoo.com (A.M.K.); 2Institute of Advanced Materials, Universiti Putra Malaysia, Serdang 43400, Malaysia

**Keywords:** recycled hematite (α-Fe_2_O_3_), ball milling technique, reduced particle size, finite element method, attenuation, complex permittivity, X-band

## Abstract

The purpose of this study was to synthesize high-quality recycled α-Fe_2_O_3_ to improve its complex permittivity properties by reducing the particles to nanosize through high energy ball milling. Complex permittivity and permeability characterizations of the particles were performed using open-ended coaxial and rectangular waveguide techniques and a vector network analyzer. The attenuation characteristics of the particles were analyzed with finite element method (FEM) simulations of the transmission coefficients and electric field distributions using microstrip model geometry. All measurements and simulations were conducted in the 8–12 GHz range. The average nanoparticle sizes obtained after 8, 10 and 12 h of milling were 21.5, 18, and 16.2 nm, respectively, from an initial particle size of 1.73 µm. The real and imaginary parts of permittivity increased with reduced particle size and reached maximum values of 12.111 and 0.467 at 8 GHz, from initial values of 7.617 and 0.175, respectively, when the particle sizes were reduced from 1.73 µm to 16.2 nm. Complex permeability increased with reduced particle size while the enhanced absorption properties exhibited by the nanoparticles in the simulations confirmed their ability to attenuate microwaves in the X-band frequency range.

## 1. Introduction

The rapid growth and diversity of applications of electromagnetic waves, particularly commercial and military electronics functioning at microwave frequencies, has attracted a lot of interest in microwave absorbing material technology for electromagnetic interference (EMI) reduction. For a microwave absorbing material to be practical, a proper balance of electrical performance, low density, thinness, mechanical properties and low cost is required. Various magnetic materials have been used in microwave absorbing applications, the most common of which are ferrites due to their excellent electrical and magnetic properties. Ferrites can be applied in various forms, such as spinels and garnets, and are often synthesized via multi-stage chemical processes, which can be complicated and expensive. Moreover, ferrites are heavy and the imaginary parts of their permittivity are very low at high frequencies, making the absorption properties dependent on magnetic loss [1]. In order to reduce the effect of these limitations, many studies have focused on new techniques such as the use of ferrites in conjunction with electrically conducting polymers [2] or dielectric carbon-based materials [3], or the doping of ferrites using various types of ionic metals [4].

Recently, Nd doped strontium ferrite was synthesized [5] using hematite (α-Fe_2_O_3_) prepared from recyclable mill scale waste. Recycled α-Fe_2_O_3_ can be produced cheaply via a simple processing technique, has low environmental waste, is stable under ambient conditions, and has unique magnetic and electrical properties. The imaginary permittivity part is, however, very low and an improvement in this property, while retaining the magnetic attributes, could increase the microwave absorption efficiency of recycled α-Fe_2_O_3_ and serve to reduce the limitations of commonly used ferrites. 

The interaction between a dielectric material and high frequency electromagnetic energy can be expressed by the relative complex permittivity equation ε* = ε′ − jε″, where ε′ and ε″, respectively, represent the real and imaginary parts. The ratio of tan δ = ε″/ε′ is the loss tangent of a material and higher values indicate higher attenuation properties. A material’s complex permittivity can be measured by a variety of methods such as the open-ended coaxial (OEC) probe technique, which is based on high frequency, room temperature, broadband measurements. The application of the OEC probe technique for the measurement of the complex permittivity of powdered and compressed materials was reported by Refs. [6,7]. However, the measurement of the complex permittivity of recycled α-Fe_2_O_3_ using the OEC technique has yet to be investigated. 

The aim of the current study was to improve the complex permittivity properties of recycled α-Fe_2_O_3_ by reducing the particle size, to nanosize, via high energy ball milling (HEBM) for several hours. The resultant effects of reduced particle size on the magnetic properties of the recycled α-Fe_2_O_3_ were also investigated. The dielectric characterization of the samples was conducted using the OEC probe technique while the magnetic properties were determined using the waveguide measurement technique. The finite element method (FEM), using COMSOL Multiphysics, was used to numerically calculate the transmission coefficients of the recycled α-Fe_2_O_3_ using microstrip model geometry. The attenuation characteristics, due to material absorption by the recycled α-Fe_2_O_3_ particles, were subsequently analyzed using calculated transmission coefficient (dB) values and FEM simulations of the electric field distributions of the microstrip covered by the recycled α-Fe_2_O_3_ particles. All measurements and simulations were performed in the X-band (8–12 GHz) microwave frequency range.

## 2. Materials and Methods 

### 2.1. Synthesis of α-Fe_2_O_3_ Nanoparticles from Mill Scale 

As illustrated in Figure 1, the mill scale flakes were crushed into coarse powder and purified, using the magnetic separation methods described in Refs. [5,8], to obtain the magnetic wustite (FeO) slurry that was then filtered and dried in a Memmert drying oven for 24 h at 30 °C. The dried FeO was then oxidized in a Protherm furnace at 600 °C for 6 h (holding time) to produce the α-Fe_2_O_3_ powder. The recycled α-Fe_2_O_3_ powder was milled, separately for 8, 10 and 12 h, into nanoparticles, at room temperature, using the SPEX Sample Prep 8000D high energy ball mill (SPEX SamplePrep LLC, Metuchen, NJ, USA), operated at 1425 rpm by a 50 Hz motor at a clamp speed of 875 cycles/min using a powder-to-ball ratio of 1:5. The steel vials containing the materials were opened for 2 min after every 50 min of milling in order to avoid the transformation of the α-Fe_2_O_3_ to Fe_3_O_4_ reported in the study by Ref. [9]. 

### 2.2. Microstructural Characterization

The phase composition structure and crystallite size of the recycled α-Fe_2_O_3_ particles before and after high energy ball milling were analyzed, using X-ray diffraction (XRD), on a fully automated Philips X’Pert High Pro Panalytical (Model PW3040/60 MPD, Amsterdam, Netherlands) with Cu-Kα radiation operating at a voltage of 40.0 kV, a current of 40.0 mA and a wavelength of 1.5405 Å. The diffraction patterns were recorded in the 2*θ* range of 2° to 80° with a scanning speed of 2°/min. All data were subjected to the Rietveld analysis on PANalytic X’Pert Highscore Plus v3.0 software (PANalytical B.V., Almelo, Netherlands). The samples were identified by comparing their diffraction peaks with those in the Inorganic Crystal Structure Database (ICSD). The average crystallite sizes were estimated based on the Scherrer formula:(1)$$D=\frac{k\lambda }{B\cos\theta },$$
where *D* is the crystallite size, *B* is the full width at half maximum (FWHM) of the diffraction peaks in radians, *k* = 0.9, *θ* is the peak position and *λ* = 1.5405 Å.

The elemental composition of the recycled α-Fe_2_O_3_ particles was analyzed using energy dispersive X-ray spectroscopy (EDX), while the size, shape and arrangement of the nanoparticles were studied using the JEM-2100F high resolution transmission electron microscope (HRTEM, JEOL, Tokyo, Japan). Drops of the α-Fe_2_O_3_ particles were dispersed in acetone, placed on copper HRTEM grids and dried. The dried materials were then transferred into the high vacuum chamber of the microscope for the viewing and analysis of the particles. The particle size distributions of the samples were obtained after processing the images, using the ImageJ software (NIH, University of Wisconsin, Madison, WI, USA), of 100 particles from each sample. 

### 2.3. Measurement of Complex Permittivity

Measurements of the real and imaginary parts of the permittivity of the recycled α-Fe_2_O_3_ particles were taken at room temperature using the Agilent 85070B open ended coaxial probe connected, via a high accuracy coaxial cable, to the Agilent N5230A PNA-L vector network analyzer (VNA, Agilent Technologies, CA, USA) at 8 to 12 GHz. A one-port reflection-only calibration was performed using a 3.5 inch high density shorting block and deionized water at 25 °C. The calibration was verified by measuring the permittivity values for unfilled polytetrafluoroethylene (PTFE) and comparing the results with the manufacturer’s values—good agreement confirmed the accuracy of the permittivity characterizations. Without a binding material, the recycled α-Fe_2_O_3_ samples were separately compressed to a thickness of 6 mm, in sample holders at 4 tons, with a mechanical hand-operated pressing machine, in order to remove air-filled gaps between the particles likely to affect the results. As shown in Figure 2, the OEC probe was then firmly placed onto the flat surface of the powdered samples for determination of complex permittivity using the equipment’s installed software.

### 2.4. Measurement of Complex Permeability

The real (μ′) and imaginary (μ″) parts of the permeability of recycled α-Fe_2_O_3_ particles were determined by the rectangular waveguide (RWG) technique. The powders were compressed into sample holders (length = 22.0 mm, width = 11.0 mm, height = 6.0 mm) and inserted into a measurement system consisting of a pair of RWGs connected to an Agilent N5230A PNA-L network analyzer. The setup was calibrated by electronic calibration modules (N4694-60001), after which measurements were made in the 8–12 GHz range based on the poly reflection/transmission precision measurement model.

### 2.5. Material Absorption Properties Based On FEM

A transmission coefficient (S_21_) is a transmission line parameter often associated with the transmission of electromagnetic waves in microwave networks and can directly describe the attenuation characteristics of a material for microwave absorption applications. In this work, the magnitudes of the transmission coefficient (|S_21_|) were theoretically calculated using FEM implemented on COMSOL Multiphysics^®^ version 3.5. The calculations were based on the model geometry of a microstrip that consisted of a dielectric (RT duroid 5880) substrate with a length of 6.0 cm, width of 5.0 cm and thickness of 0.15 cm and with a signal line (width = 1.5 mm, length = 6.0 cm) etched on the surface of the substrate along the broader side. The complex permittivity of the substrate was 2.2 − j·0.00088. The input and output ports comprised radio frequency (RF) subminiature connectors attached to both ends of the substrate with the inner conductor of each subminiature placed in contact with the signal line of the microstrip. The procedures for the FEM analysis consisted of: (a) discretizing the solution region into elements of a finite number of sub-regions, (b) deriving the central equations of a typical element, (c) assembling the elements in the region of the solution, and (d) solving the obtained system of equations. The measured complex permittivity values of the recycled α-Fe_2_O_3_ particles were used as inputs for the calculations in the frequency range of 8–12 GHz (X-band). RF electromagnetic waves, in harmonic propagation mode, were applied to the model for FEM and solved, by solving Maxwell’s equations for a typical tetrahedral element/mesh (Figure 3), to determine the values of the transmission coefficients of the samples. The attenuation of the material absorption was then deduced from the calculated |S_21_| based on the following formula:Transmission coefficient magnitude (dB) = 20log (|S_21_|)(2)

## 3. Results and Discussion

### 3.1. Microstructural Characterization

The X-ray diffractograms of the recycled α-Fe_2_O_3_ particles before and after 8, 10 and 12 h of ball milling are shown in Figure 4. The diffractograms were compared with the standard patterns of the Inorganic Crystal Structure Database (ICSD) and all the Bragg peaks were identified as single phase rhombohedral (hexagonal) crystal structures of α-Fe_2_O_3_ with an R-3c space group. There were no other phases identified, which was consistent with similar work in which recycled α-Fe_2_O_3_ was synthesized from mill scale [10]. This suggests that α-Fe_2_O_3_ did not transform to Fe_3_O_4_ during the ball milling since both the unmilled and milled particles possessed identical crystal structures. It was also observed that the maximum intensity peaks of the milled particles were, unexpectedly, located on the (110) plane, while, for the unmilled particles, the maximum intensity peak was on the (104) plane. This could be attributed to preferred orientation [11], magnetic ordering [12] or an increase in crystallinity as a result of the ball milling. We also observed that as the milling time increased, the peaks broadened and the sharpness decreased when the crystallite size decreased [13]. The ICSD reference numbers for the α-Fe_2_O_3_ powders were 98-002-2616, 98-100-2733, 98-004-0652 and 98-009-4106 for the unmilled and 8, 10 and 12 h milled particles, respectively. 

The lattice parameters *a*, *b* and *c* and the average crystallite size *D* were calculated after performing the Rietveld refinement of the peak profiles of the XRD analysis of the recycled α-Fe_2_O_3_ particles. As depicted in Table 1, the lattice parameter *a* increased fractionally when the crystallite size decreased and as the milling time increased. Lattice parameter expansion with reduced crystallite size in α-Fe_2_O_3_ nanoparticles was reported to indicate the surface disorder of the nanograins [14] as a result of mechanical milling. Lattice parameter expansion has also been related to the large surface-to-volume ratio of the nanoparticles, which contributes to the relaxation of the lattice vibration leading to the expansion [15]. Overall, the cause of the lattice expansion of nanocrystalline oxides has been mainly attributed to lattice defect formation due to oxygen vacancy [16]. This observation possibly implies the presence of interfacial structural disorder and the formation of oxygen vacancies in the recycled α-Fe_2_O_3_ nanoparticles. This is in agreement with a previous study of commercial α-Fe_2_O_3_ nanoparticles prepared with the ball milling technique [17].

The spectra of the EDX analysis of the recycled α-Fe_2_O_3_ particles are shown in Figure 5 and the elemental compositions are summarized in Table 2. The results showed that the unmilled particles were composed of 98.67% Fe and O with about a 1% impurity in the form of Mg and Mn elements. After 8, 10 and 12 h of milling, the Fe and O content increased, respectively, to 99.79%, 99.83% and 99.85% with very low (less than 0.21%) levels of Ca and Mg impurity. Similar results, reported by Nadhirah et al. [18], confirmed that the oxidized mill scale waste formed a high percentage of α-Fe_2_O_3_ along with a small amount of impurity compounds. The results indicated that the recycled α-Fe_2_O_3_ nanoparticles prepared using the ball milling technique were high purity. 

Figure 6 shows the HRTEM micrographs of the microstructure of recycled α-Fe_2_O_3_ particles at various stages of milling. We observed that before milling the particles were randomly sized, bulky, loosely formed and not showing any distinct aggregation. However, as the milling progressed, the particles got work hardened, leading to their refinement into very fine nanoparticles with noticeable aggregation and agglomeration. The increase in agglomeration can be attributed to an increase in specific surface area, associated with reduced particle size, resulting in higher adhesion forces and more aggregation [19]. The nanoparticles appeared to be largely spherical in shape and their particle size distribution, as displayed in Figure 7, generally indicated a refinement and reduction in size with increased milling time. The particles of the unmilled recycled α-Fe_2_O_3_ were in the range 0.8–3.0 µm with an average size of 1.73 µm. After 8, 10 and 12 h of ball milling, the reduced particle sizes were, respectively, in the ranges 11.5–34.6, 11.0–28.6 and 10.3–24.5 nm. These results are in close agreement with the average crystallite sizes estimated by XRD measurement. The average particle sizes were calculated to be 21.5, 18 and 16.2 nm for the 8, 10 and 12 h of milling time. 

### 3.2. Dielectric Characterization

The frequency variation in ε′ and ε″ for recycled α-Fe_2_O_3_ with particle sizes of 16.2, 18, and 21.5 nm and 1.73 µm are presented in Figure 8. The results showed that the within the frequency range the real and imaginary parts of permittivity increased significantly when the particle size was reduced, to nanosize, through ball milling. At 8 GHz, ε′ increased from 7.617 to 12.111, while ε″ increased from 0.175 to 0.467 when the particle size was decreased from 1.73 µm to 16.2 nm. Therefore, the sample with the smallest particle size had the highest complex permittivity values. This represents an improvement in the complex permittivity of recycled α-Fe_2_O_3_ particles with reduced particle sizes. It is also evident, Figure 9, that the loss tangent increased moderately when the particle size decreased in the frequency range, which indicated an enhancement in the attenuation properties of the recycled α-Fe_2_O_3_ as the particle sizes got smaller. 

The increase in the complex permittivity can be related to the enhancement of interfacial polarization due to the effect of interfacial lattice disorder and the oxygen ion vacancies formed during ball milling, which increased with reduced particle size in agreement with the work of Reference [12]. Additionally, smaller particles were more compact, had less air gaps and, therefore, formed good contact between the constituent particles leading to the observed increase in the density of the interfaces. The resulting interfacial polarization, therefore, increased, which led to the overall increase in dielectric permittivity. In general, both the real and imaginary parts of permittivity were found to be high, at 8 GHz, which decreased with a further increase in frequency to 12 GHz for all the particle sizes. This was consistent with the behavior of the dielectric properties of ferrites with frequency [20,21]. The decrease in ε′ with the increase in frequency was attributed to the electron hopping between Fe^3+^ and Fe^2+^ ions whose frequency follows the applied electric field, resulting in the increase in ε′ at low frequency. However at high frequencies, this hopping frequency lagged behind the applied electric field causing a decrease in ε′ due to disorderly dipolar orientation [22]. 

The imaginary part of permittivity represented the loss of electrical energy. At low frequencies, the grain boundaries had high resistivity and, because of the buildup of electrons at the grain boundaries, a higher energy acquisition was required for electron hopping leading to a higher loss of electrical energy. Conversely, at high frequencies the grain boundaries had low resistivity and, therefore, less electrical energy loss from electron hopping [21,23].

### 3.3. Complex Permeability

The effect of the milling treatment on the magnetic properties of the recycled α-Fe_2_O_3_ particles was investigated using complex permeability measurements. The variations of the real (μ′) and imaginary (μ″) parts with frequency, due to the different particle sizes, are shown in Figure 10. It is evident in Figure 10 that both μ′ and μ″ increased significantly with reduced particle size within the frequency range of measurement. Moreover, μ′ decreased with frequency and varied between 0.90 and 1.45 while μ″ increased with frequency reaching a maximum value of 0.161 at 11.2 GHz. The increase in permeability with reduced particle size may be attributed to the observed preferred orientation, leading to magnetic ordering as well as surface effects [12] in the microstructure of the milled particles and enhanced magnetization.

### 3.4. Material Absorption

Figure 11 presents the variation of simulated |S_21_| (dB) with frequency and clearly illustrates the effect of the reduced particle sizes on the attenuation characteristics of the recycled α-Fe_2_O_3_ particles as a result of absorption. The profiles indicated that the smaller the particle size, the lower the |S_21_| (dB) values, which implied higher attenuation from material absorption, consistent with Figure 9. The simulated |S_21_| depended on the inputs of the complex permittivity values and, therefore, the higher ε″ values of the smaller α-Fe_2_O_3_ particles could have resulted in the higher absorption of electromagnetic energy resulting in lower |S_21_| (dB). Additionally, the |S_21_| (dB) of the particles decreased with frequency throughout the measurement range due to the effects of skin-depth [24].

FEM simulations of the x-component of the electric field (V/m) distribution at 12 GHz of the microstrip covered with the recycled α-Fe_2_O_3_ particles can be visualized as depicted in Figure 12. It is evident from Figure 12 that the electric fields decreased from the input port to the output port as the particle size was reduced. The reduction in the electric field distribution, with reduced particle size, is in agreement with the simulated |S_21_| (dB) since the smaller particles possess higher ε″ leading to higher attenuation due to more absorption of the microwaves. 

## 4. Conclusions

The complex permittivity of high quality α-Fe_2_O_3_ particles recycled from mill scale waste was enhanced significantly after reducing the particles to nanosize via high energy ball milling for several hours. The milling treatment produced an equal increase in permeability and reduction in particle size. This positioned the recycled α-Fe_2_O_3_ nanoparticles as potential magneto–dielectric microwave absorbing materials. Microstructural defects and disorders, which occurred in the milled particles as the sizes reduced, facilitated the improvements in the electromagnetic properties of the recycled α-Fe_2_O_3_ nanoparticles. The simulated absorption properties exhibited by the nanoparticles confirmed their ability to attenuate microwaves in the X-band frequency range. The significant electromagnetic properties of recycled α-Fe_2_O_3_ linked to the size of the nanoparticles can be employed in possible applications requiring tunable attenuation of electromagnetic energy. Recycled α-Fe_2_O_3_ nanoparticles with improved dielectric and magnetic properties are cheaper to produce, lighter, less environmentally wasteful, and could reduce the limitations associated with the ferrites commonly used in microwave absorption applications.

## Figures and Tables

**Figure 1 materials-12-01696-f001:**
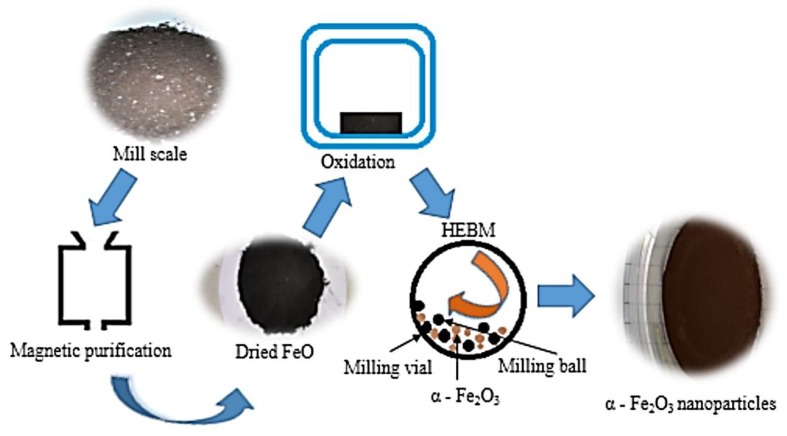
Synthesis of recycled α-Fe_2_O_3_ nanoparticles.

**Figure 2 materials-12-01696-f002:**
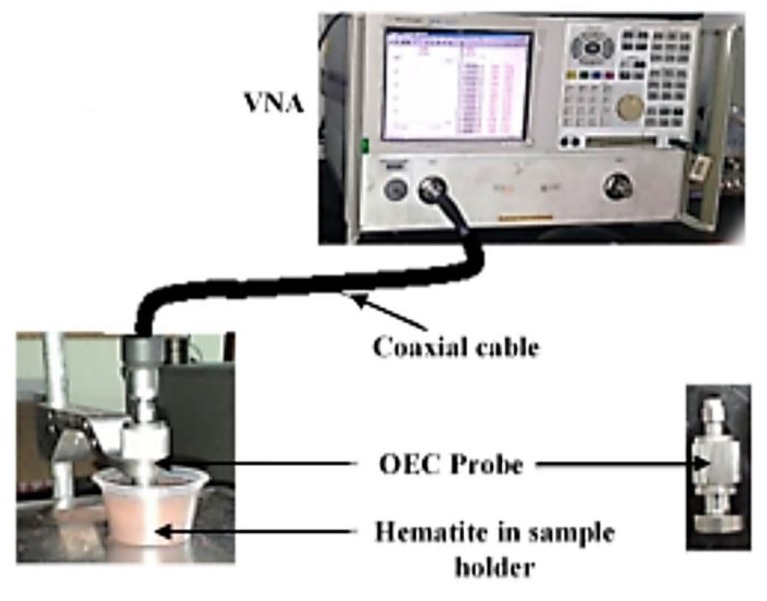
Measurement of complex permittivity using OEC technique.

**Figure 3 materials-12-01696-f003:**
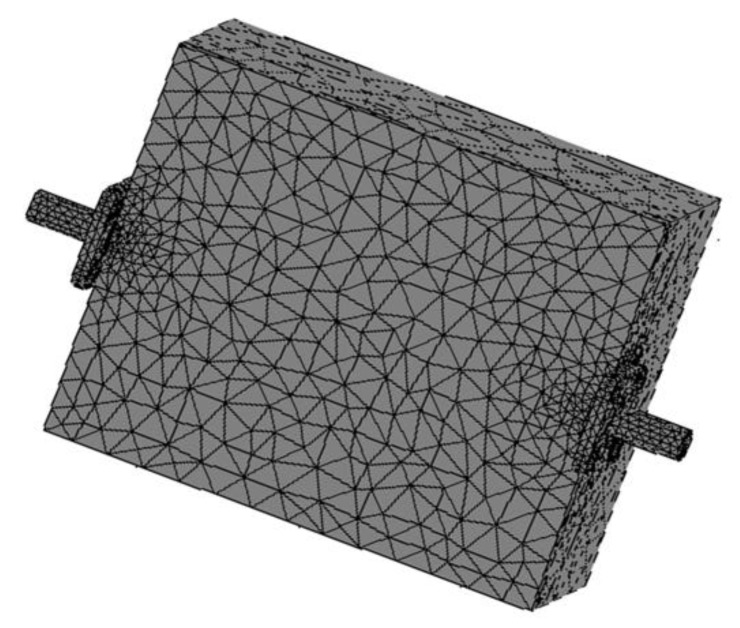
Mesh of microstrip for FEM simulation.

**Figure 4 materials-12-01696-f004:**
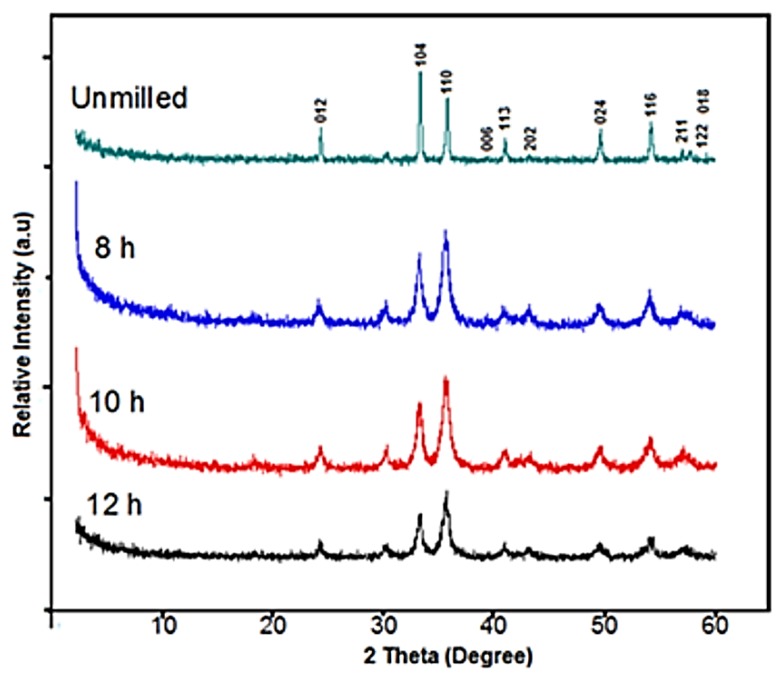
XRD diffractograms of the recycled α-Fe_2_O_3_ particles.

**Figure 5 materials-12-01696-f005:**
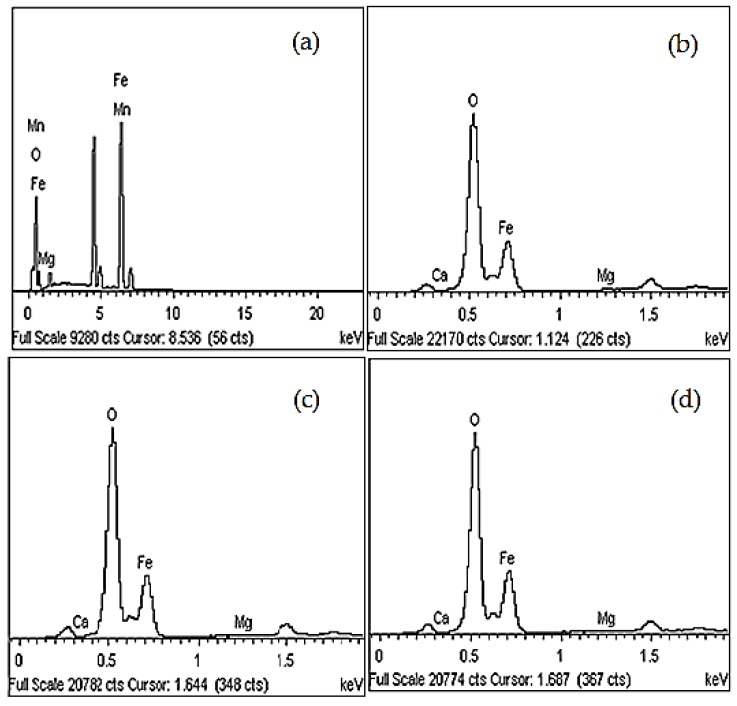
EDX spectra of recycled α-Fe_2_O_3_ after (**a**) 0 h, (**b**) 8 h, (**c**) 10 h, and (**d**) 12 h of milling time.

**Figure 6 materials-12-01696-f006:**
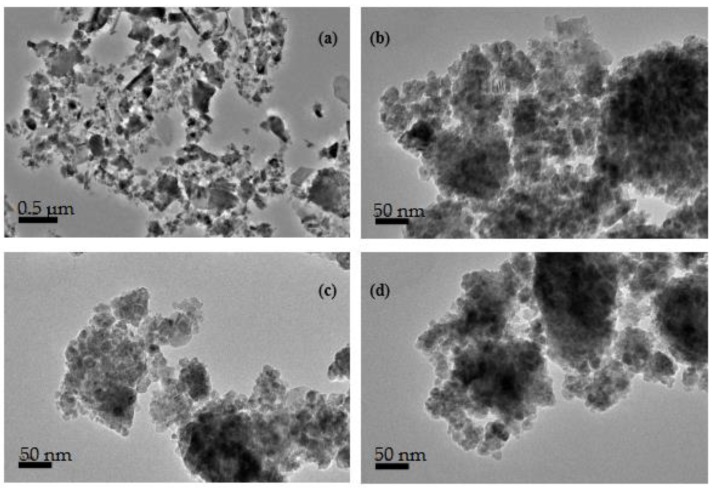
HRTEM micrographs of recycled α-Fe_2_O_3_ at (**a**) 0 h, (**b**) 8 h, (**c**) 10 h, and (**d**) 12 h.

**Figure 7 materials-12-01696-f007:**
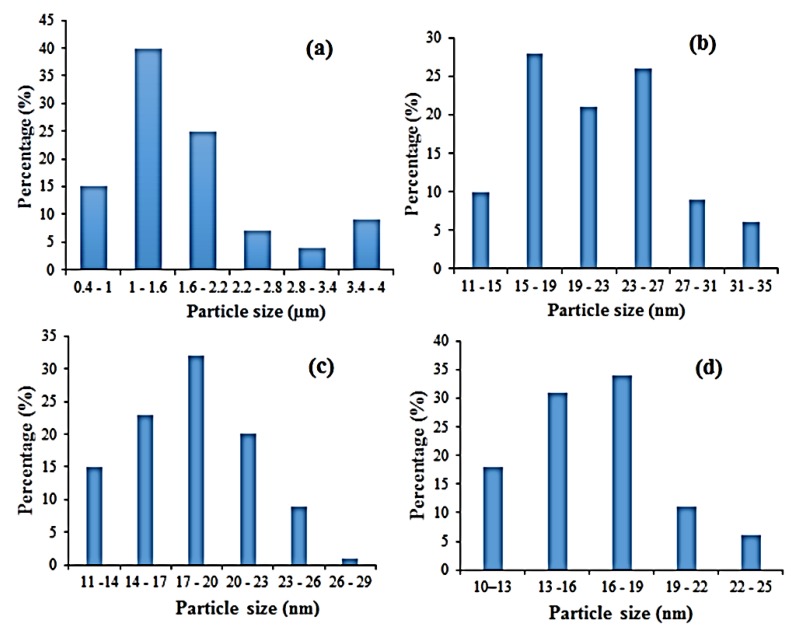
Particle size distribution of recycled α-Fe_2_O_3_ at (**a**) 0 h, (**b**) 8 h, (**c**) 10 h, and (**d**) 12 h.

**Figure 8 materials-12-01696-f008:**
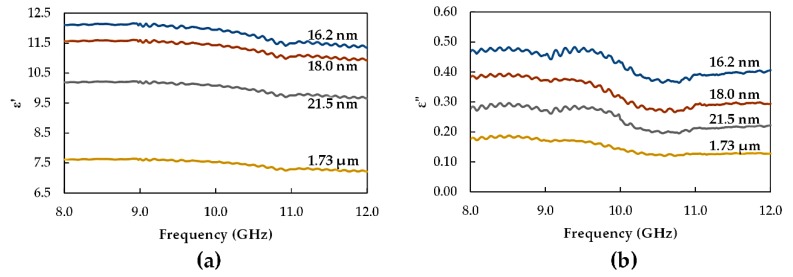
Variation of (**a**) ε′ and (**b**) ε″ with frequency for different recycled α-Fe_2_O_3_ particle sizes.

**Figure 9 materials-12-01696-f009:**
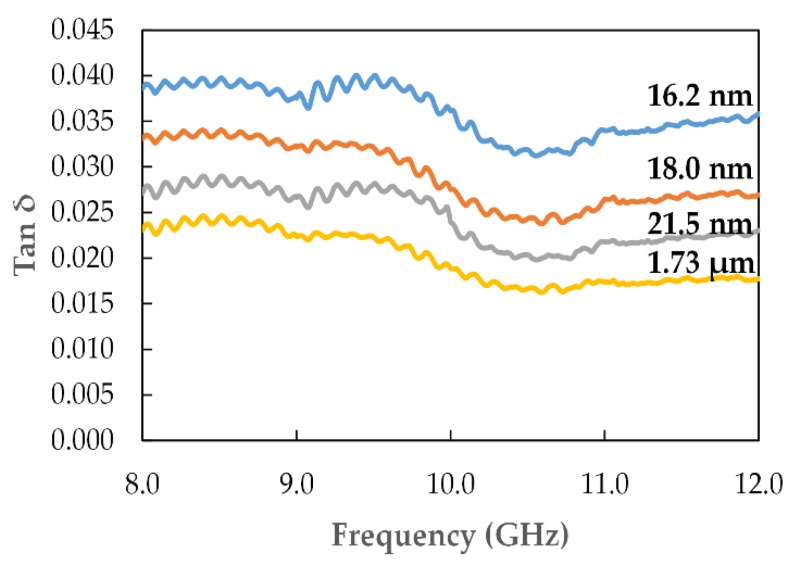
Variation in loss tangent with frequency for different recycled α-Fe_2_O_3_ particles.

**Figure 10 materials-12-01696-f010:**
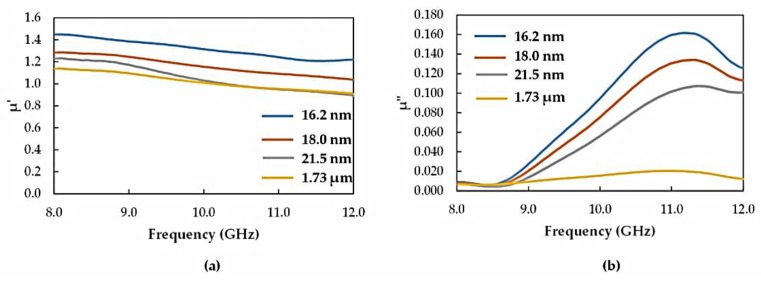
Variation of (**a**) μ′ and (**b**) μ″ with frequency for the recycled α-Fe_2_O_3_ particles.

**Figure 11 materials-12-01696-f011:**
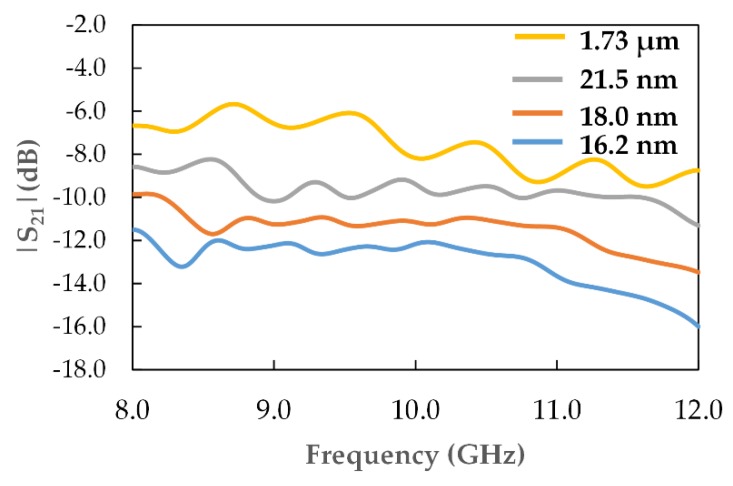
|S_21_| (dB) for different recycled α-Fe_2_O_3_ particles as a function of frequency.

**Figure 12 materials-12-01696-f012:**
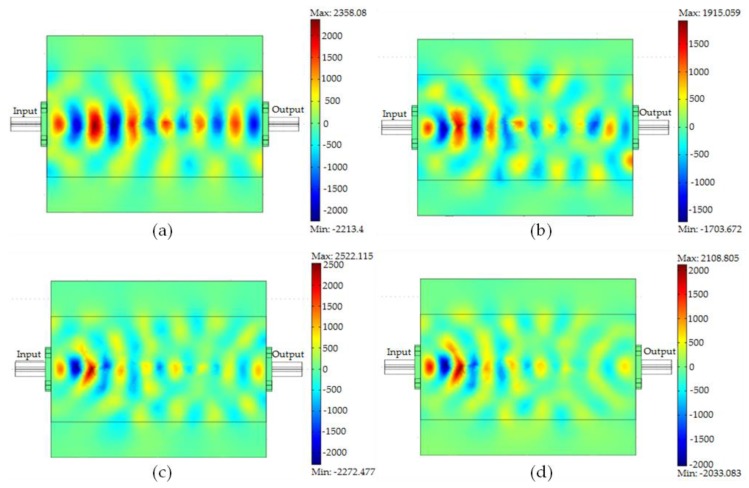
Electric field distribution patterns for recycled α-Fe_2_O_3_ particles of (**a**) 1.73 μm, (**b**) 21.5 nm, (**c**) 18.0 nm, and (**d**) 16.2 nm.

**Table 1 materials-12-01696-t001:** Lattice parameters and average crystallite size as a function of milling time.

Condition	*a* = *b* (Å)	*c* (Å)	*D* (nm)
Unmilled	5.0290	13.7360	106.2
8 h of milling	5.0340	13.7480	12.3
10 h of milling	5.0350	13.7500	11.8
12 h of milling	5.0380	13.7390	11.1

**Table 2 materials-12-01696-t002:** Quantitative analysis of recycled α-Fe_2_O_3_ particles.

Sample	Element (%)					
	Fe	O	Mn	Mg	Ca	Total
Unmilled	78.07	20.69	0.69	0.55	-	100.0
8 h of milling	58.87	40.92	-	0.11	0.10	100.0
10 h of milling	56.70	43.13	-	0.12	0.05	100.0
12 h of milling	57.22	42.63	-	0.09	0.06	100.0

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
