# Peer review of "Enhancement of Complex Permittivity and Attenuation Properties of Recycled Hematite (α-Fe2O3) Using Nanoparticles Prepared via Ball Milling Technique"

_materials, 2019, doi:10.3390/ma12101696_

Reviewer 1 Report

The paper presents the X-band MW characterization of ferrite micro/nano-powder, focusing on the effectiveness of particles size reduction by milling technique in enhancing the material EM absorption capability. Methodology and results are clearly presented, and may be of interest for the skilled reader.

By the way, the following correction/improvements are pointed out in order to consider the manuscript worth of mention for publication.

 Ferrite-based materials are currently studied for their magnetic properties, thus,  for research completeness, measurements of permeability would be appreciated. In this regard, for instance, milling has surely ‘good’ impact on permittivity: is there any variation in the magnetic behavior? (it may be the case, since, as reported in Tab.2, Fe amount becomes considerably lower due to milling treatment).

Fig.5 images quality should be improved (the magnification scale-bar value is very hardly visible in the current version).

The legend inside Fig.8 plot must be top/down reversed.

In this kind of papers, ε′ is referred to as ‘real part of permittivity’ (not as ‘dielectric constant’) and  ε″ as ‘imaginary part of permittivity’ (not as ‘loss factor’) – see for ex. Microwave electronics; measurement and materials characterization. West Sussex, England: John Wiley; 2004 – correct everywhere in the text and in Fig.7 (Y-axes tag).

Tab.3 does not provide further information with respect to Fig.7-8 plots: it should be deleted.

In this kind of analyses, when a quantity/parameter/coefficient is given/defined in dB having negative values (i.e., it is a log of a rate which ranges in 0-1), the typical meaning is that ‘close to 0 dB’ is an indication of ‘high’ (rate close to unity) while ‘below minus ten(s) dB’ is an indication of ‘low’ (rate toward zero). Thus, the word ‘Attenuation’ as given in eq.(2) may lead to confusion, that quantity being better defined as simply ‘Transmission’ (of course...is congruent to the transmission scattering parameter!), which results higher for larger size powder (lower absorption, lower attenuation) and lower for smaller size powder (higher absorption, higher attenuation); however, the two plots of Fig.9 give, obviously, the same information: one of them is absolutely useless and should be deleted. In order to improve the quality of the work presentation, Authors are suggested to referring to recent cited MW works as IEEE Transactions on Microwave Theory and Techniques 65 (2017) 2801-9 and Acta Astronautica 134 (2017) 33-40 ,where further suggestions aimed at achieve better and better performances on larger bandwidths by investigating on graded composites (i.e., by coupling different materials ) as well as on design optimization methods are retrievable.

 Author Response

List of Responses to the Reviewer’s Comments on the Manuscript

(Materials -468929)

Enhancement of the complex permittivity and attenuation properties of recycled hematite (α – Fe2O3) using nanoparticles prepared via the ball milling technique

Thank you for reviewing our submission and your very valuable comments. We carefully responded to all the comments and have modified the manuscript accordingly.

The following are our responses to the specific comments raised -

1.    For research completeness, measurements of permeability would be

 appreciated.

 Magnetic properties of the recycled α – Fe2O3 particles have been incorporated in the revised submission. Permeability measurement procedure has been described (Line 125-131) and discussed (Line 265-276).

2.    Fig.5 images quality should be improved

The Figure has been re-captioned Figure 6 and modified to show improved magnification scale bar. Line 218-219

3.    The legend inside Fig.8 plot must be top/down reversed

Legend has been top/down reversed. Figure re-captioned Figure 9. Please check line 241-242                                                                                                      

4.    ε′ is referred to as ‘real part of permittivity’ (not as ‘dielectric constant’) and ε″  

as ‘imaginary part of permittivity’ (not as ‘loss factor’). Correct everywhere in   

the text and in Fig.7 (Y-axes tag).

 ‘Dielectric constant’ and ‘loss factor’ changed to ‘real part of permittivity’ and ‘imaginary part of permittivity’ respectively throughout the revised submission. Figure 7 re-captioned Figure 8 and the Y – axes re-labelled accordingly.

5.    Tab.3 does not provide further information with respect to Fig.7-8 plots: it should be deleted.

Table 3 deleted from the revised submission

6. (a)    The word ‘Attenuation’ as given in eq. (2) may lead to confusion, that quantity being better defined as simply ‘Transmission’.

 Attenuation as represented in equation (2) has been changed to transmission coefficient (dB). Please check line 154

 (b)    The two plots of Fig.9 give, obviously, the same information: one of them and should be deleted

 Figure 9 re-captioned Figure 11. The Y-axis re-labelled ‘|S21| (dB)’. Please check line 291-292.

 (c)    Authors are suggested to referring to recent cited MW works as IEEE Transactions on Microwave Theory and Techniques 65 (2017) 2801-9 and Acta Astronautica 134 (2017) 33-40.

The recent works stated were reviewed by the authors for the improvement of the quality of our revised submission. Relevant portions have been appropriately cited.

 • All the comments made by the reviewer have been acted upon.                         

Thank You.

Ebenezer Ekow Mensah

Physics Department, Faculty of Science, Universiti Putra Malaysia.

([email protected])

 Reviewer 2 Report

The reviewer has failed to find scientific investigation in this article. The text describes characteristics of samples, which are different to each other only by the time of milling. The main idea of the text is: "the longer milling - the better properties", but, unfortunately, it is not a research. The research means some new technological approaches, new properties, new explanations, anything new. In this text the reviewer has seen only the characterization of samples by different methods and nothing new.

Author Response

List of Responses to the Reviewer’s Comments on the Manuscript

(Materials -468929)

Enhancement of the complex permittivity and attenuation properties of recycled hematite (α – Fe2O3) using nanoparticles prepared via the ball milling technique

 Thank you for reviewing our submission and your very valuable comments. We carefully responded to all the comments and have modified the manuscript accordingly.

The following are our response to the comments raised -

The reviewer has failed to find scientific investigation in this article. The text describes characteristics of samples, which are different to each other only by the time of milling. The main idea of the text is: "the longer milling - the better properties", but, unfortunately, it is not a research. The research means some new technological approaches, new properties, new explanations, anything new. In this text the reviewer has seen only the characterization of samples by different methods and nothing new.

We have acted on your valuable comments and highlighted the following in our revised submission.

1.      The potential use of recycled α – Fe2O3 nanoparticles with improved electromagnetic properties as a substitute to the most commonly used magnetic fillers for microwave absorption applications.

 2.      Recycled α – Fe2O3 nanoparticles with improved electromagnetic properties are cheap to synthesize from industrial waste materials and save the environment from environmental pollution.

 3.      The use of milling treatment to tune the electromagnetic properties of recycled α – Fe2O3 nanoparticles.

 Thank You.

Ebenezer Ekow Mensah

Physics Department, Faculty of Science, Universiti Putra Malaysia.

([email protected])

 Round  2

Reviewer 1 Report

Authors have revised the paper successfully: it can be now considered for publication in the present form.

Reviewer 2 Report

The authors have made the valuable revision of the text. In present form it can be published in Materials.